# Parental non-involvement strategy for handling sibling conflict on social avoidance in migrant children: Chain mediation of sibling conflict and parent-child conflict

**Yuge Yue[1], Kaiwu Lu[2]\*, Danping Ye[3]**

1 Faculty of Education, Ningde Normal University, Ningde, Fujian, People's Republic of China, 2 Academic Affairs Office, Ningde Normal University, Ningde, Fujian, People's Republic of China, 3 Ningde Institutional Kindergarten, Ningde, Fujian, People's Republic of China

\* yyg310@126.com

**Data Availability Statement:** All relevant data are within the manuscript and its Supporting Information files.

## Abstract

In the process of urbanization, the social adaptation of migrant children has become an important issue in their development. This study adopts family systems theory and ecological systems theory to examine the effects of parental non-involvement strategies in handling sibling conflict on migrant children's social avoidance. It also investigates the mediating role of sibling conflict and parent-child conflict. The results of the study, reported by parents of 253 mobile children with siblings, suggest that parental strategies of not intervening in sibling conflict are an important factor influencing the development of social avoidance in mobile children. The Parental strategy of not intervening in sibling conflict had an effect on migrant children's social avoidance through the separate mediating effect of parent-child conflict, and also through the chained mediating effect of sibling conflict and parent-child conflict. The study also found that the separate mediating effect of sibling conflict was not significant. This study contributes to the research on the relationship between parental non-intervention in sibling conflict and migrant children's social avoidance. It also highlights the impact of sibling conflict and parent-child conflict on migrant children's social avoidance by establishing and validating a comprehensive research model. The results of the study can help parents establish close parent-child relationships for migrant children and provide scientific guidance for children to develop positive sibling relationships. This, in turn, can assist migrant children in better adapting to a new social environment.

## 1 Introduction

Social withdrawal behavior refers to a child's behavior that inhibits his or her participation in peer interactions and exhibits solitude in social situations [1]. According to social motivation theory [2], children's social withdrawal behaviors are determined by a combination of two dimensions: social tendency motivation and social avoidance motivation. These dimensions can be further classified into three subtypes: shyness, social apathy, and social avoidance. Social

**Funding:** This study was supported by grants from the Office for Philosophy and Social Sciences in Fujian Province, People's Republic of China, Title: "Examining Family Relationships in the Urban Floating Population of Fujian Province in the New Era" (FJ2022X022), and the Collaborative Innovation Project in Ningde Normal University, Fujian, People's Republic of China, Title: "Collaborative Innovation Center for Cultivating Excellent Early Childhood Teachers" (2023ZX04).

**Competing interests:** The authors have declared that no competing interests exist.

avoidance is the most adaptively risky of the three subtypes of social adjustment [3] and refers to a type of prosocial behavior in which children prefer to be alone, avoid a variety of social situations, and have low convergent and high avoidance motivations for social behavior [4]. The negative impact of social avoidance on children's social adjustment has been supported by numerous national and international studies. For example, an Indian study on adolescent social adjustment found that social avoidance was significantly and positively associated with peer rejection and loneliness [5]. Studies in China have also found that social avoidance in kindergarten and first-grade children significantly predicts children's later peer problems and depression [6]. Chinese kindergarten and first-grade children also hold the most negative attitudes toward socially avoidant children [7]. Not only that, but there are also studies showing that peers have more negative attitudes towards socially avoidant children. For example, Italian preschoolers generally perceive socially avoidant children as having lower IQ and poorer relationships with their teachers [8]. After the implementation of China's "individual two-child" and "comprehensive two-child" family planning policies in 2016, the family structure has changed, and multi-child families have become the mainstream of China's family model. In particular, the children of urban migrant workers (referred to as migrant children) have to adapt not only to the arrival of their compatriots but also to the new social environment. Therefore, a study on the social adaptation of urban migrant children will help them avoid social isolation and thus better adapt to the new social environment.

It has been suggested that social avoidance is an extreme form of shy withdrawal [9], and that shyness is associated with a range of negative adaptations such as peer relationship difficulties, internalizing problems, low self-esteem, and poorer academic performance [10–13]. There is also research suggesting that social avoidance stems from one's own feelings of depression [4], and some studies focus on the importance of experiences in children's development. It is suggested that negative experiences in childhood, such as rejection, bullying, and conflicts with peers, can cause children to avoid or reject social interactions with others [5,14]. Studies have examined the causes of social avoidance from both endogenous factors (such as depression) and exogenous factors (such as growth experiences). This is in line with Bronfenbrenner's ecological systems theory [15], which suggests that children's social adaptation is influenced by various factors, including the individual themselves and their living environment. This study will examine the mechanisms of family, sibling, and other influences on social avoidance in migrant children.

## 1.1 Non-involvement strategy and social avoidance

Siblings, a result of the two-child birth policy, have become significant figures in children's development. The interactions and exchanges children have with their siblings as they mature have evolved into crucial developmental experiences for them. Research has shown that when siblings spend more time together, there is more conflict between them [16]. Handling conflicts among children has become an important topic for parents to learn about. Parents' non-involvement strategy for handling sibling conflict refers to a coping mechanism in which parents adopt a non-involvement strategy towards their children's disputes. They may choose to ignore or refrain from intervening, allowing the children to resolve the conflict independently [17]. For young children who are unable to resolve conflicts independently, this non-intervention strategy is a negative response driven by parental avoidance. Children who grow up with negative parenting styles are prone to anxiety and withdrawal during social interactions [18], exhibit more implicit and explicit problem behaviors, and experience poorer social adjustment [19,20]. Research in China has also found that children's social adjustment improves with more positive parenting styles. Conversely, the more negative parenting styles

are, the worse the children's social adjustment is, and can lead to an increase in children's disciplinary aggression and shyness-anxiety behaviors, and a decrease in children's sensitive cooperative behaviors [21].

Therefore, we hypothesized that,

H1: Parental non-intervention in sibling conflict was significantly and positively associated with children's social avoidance.

## 1.2 The mediating role of sibling conflict

Sibling conflict is a significant factor in the development of individuals and can have adverse effects on individual growth. It can lead to the development of sleep problems and social behavioral issues, including behavioral disorders and difficulties in peer interactions [22–24]. Conflicting sibling relationships have also been associated with more emotional problems, academic adjustment issues, peer rejection, and impulsive behaviors [25,26]. Based on Situational Experience Theory, research has shown that negative experiences, such as bullying and rejection, diminish children's inclination to engage socially, leading to social avoidance behavior [5,14]. A two-year follow-up study of adolescents found that sibling aggression (physical and verbal) significantly predicted their roles as bullies and victims in peer relationships [27]. It can be inferred that sibling conflict may lead to social avoidance behavior in children.

Some studies have found that parents' non-involvement strategy in sibling conflicts positively predicts the occurrence of such conflicts. The higher the level of parental non-involvement strategy for sibling conflict, the higher the level of competition and confrontation in the sibling relationship, resulting in increased sibling conflict [28].

The impact of parental non-intervention in sibling conflict has also been validated in Chinese studies: uninvolved siblings have the lowest closeness scores and relatively high conflict scores [29]. The more passive the non-involvement, the greater the level of confrontation and competition in young children's sibling relationships [30] Other studies have explored the significance of parental involvement in resolving sibling conflicts, considering variables such as the siblings' ages and their conflict resolution abilities. They suggest that parents' non-intervention strategies are more appropriate for adult intersibling conflict because adults already possess the ability to handle and resolve conflicts between themselves and their siblings. Conversely, parents' non-intervention strategies exacerbate sibling conflict among younger children [31]. Family systems theory [32] views the family as an interconnected and interacting dynamic system in which any subsystem has an impact on other systems. It has been confirmed that parents' non-involvement strategy affects the development of sibling relationships. Non-involvement strategy leads to an increase in sibling conflict, which, in turn, increases the likelihood of children's social avoidance.

Therefore, this study hypothesizes that,

H2: Parental nonintervention in sibling conflict strategies can have an effect on social avoidance through the mediating role of sibling conflict.

## 1.3 The mediating role of parent-child conflict

The parent-child relationship is crucial in family dynamics. It affects not only the development of individual children but also the construction of their other social relationships. It has been found that in the process of constructing good parent-child relationships, children acquire basic knowledge, skills, behaviors, and values, and these qualities are important factors that influence the development of children's other social relationships [33–35]. Poor parent-child

relationships can affect children's mental health and lead to problems such as behavioral disorders, psychosis, suicidal and criminal tendencies [36]. Children in negative, conflictual parent-child relationships exhibit more disruptive and aggressive behavior [37,38]. Because the early mother-infant relationship prospectively predicts child behavioral problems[39], a conflicted mother-child relationship predicts aggression, disciplinary problems, and anxiety in children [40]. Similarly, a conflicting father-child relationship is an important factor influencing children to develop adjustment problems [41].

It has also been found that the conflictual nature of the mother-child relationship is usually linked to the mother's rejecting behavior towards the child. Frequent neglect of the child's needs, showing a lack of sensitivity and concern for the child, can result in a lack of emotional support for the child, leading to coldness and indifference in the parent-child relationship [42]. Negative parenting behaviors such as insensitive and less responsive mothers can predict subsequent adjustment problems in children [43]. Parenting behaviors such as threatening, coercive, controlling, corporal punishment, anger, scolding, and the use of violent, aggressive words and actions are detrimental when frequently directed towards children. These behaviors can lead to tension and conflict in the parent-child relationship [44]. Sibling conflict is a common phenomenon in families with multiple children. How parents handle sibling conflict and respond to issues that arise between children can affect not only the parent-child relationship but also children's social behavior.

It is hypothesized that,

H3: Parent-child conflict has a mediating role in parental non-intervention in sibing conflict and children's social avoidance.

## 1.4 Chain mediation of sibling conflict and parent-child conflict

Studies have found that the strategy of parental non-intervention is detrimental to children's ability to learn communication skills in the midst of conflict. This approach also causes children to miss out on opportunities to acquire conflict management strategies, which may exacerbate the conflict when sibling conflict reoccurs due to the child's lack of appropriate conflict resolution [28]. The Chinese study also found that mothers' strategies for handling sibling conflict were significantly and positively related to young children's sibling relationships [45], and thus it can be hypothesized that parental nonintervention is associated with more sibling conflict. One study found that sibling conflict was related to both father-child and mother-child conflict [46]. Inter-sibling squabbling and discord can cause stress and worry for parents, which can affect their attitudes toward their children, leading to a shift toward harsh, authoritarian parenting or alienation [47]. Parenting styles that are harsh, authoritarian, or detached are not conducive to the development of self-control in children [48]. This lack of self-control can lead to difficulties in managing their emotional and behavioral responses [49]. Children with low self-control tend to exhibit more aggressive behaviors, experience more interpersonal conflicts, and are more susceptible to both implicit and explicit problem behaviors [50–52]. It has also been found that long-term parent-child conflict can lead to many social problems in individuals, such as negative social adaptation [53,54], anxiety [55,56], and depression. Another study also found that parent-child conflict was significantly and positively related to social avoidance [57].

Therefore, this study hypothesized that parental non-intervention in sibling conflict would lead to increased sibling conflict and increased parent-child conflict. This, in turn, would result in more social adjustment problems in migrant children. It is hypothesized that,

H4:Sibling Conflict and Parent-Child Conflict Chain Mediated Between Parental Nonintervention in Sibling Conflict Strategies and Social Avoidance in Mobile Children.

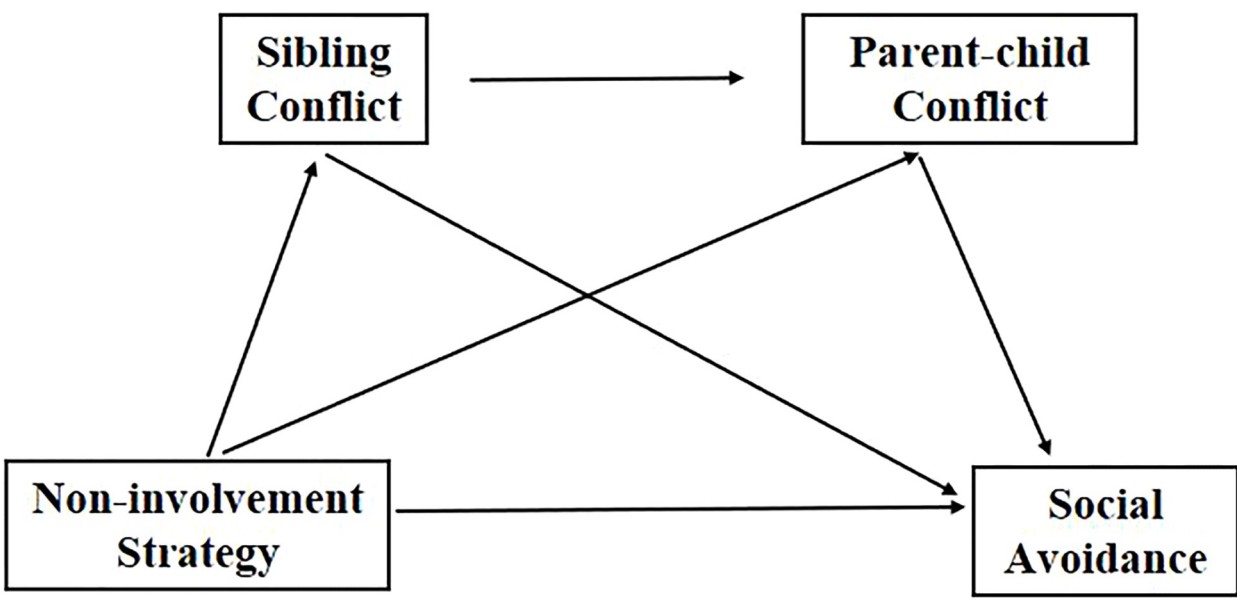

**Fig 1. Chain-mediated model.**

Based on the above hypotheses, the proposed influence mechanism of sibling conflict and parent-child conflict is illustrated in Fig 1.

## 2 Research methodology

### 2.1 Data collection

After receiving approval from the Research Ethics Review Committee of Ningde Normal University, this study used convenience sampling to select 10 kindergartens in the four cities of Xiamen, Zhangzhou, Quanzhou, and Ningde in China Fujian Province. Subsequently, 30 children with siblings were randomly chosen from each kindergarten, totaling 300 children. With the assistance of kindergarten teachers, the children's birthplaces were verified, leading to the exclusion of 42 local children. Finally, 258 migrant children were included in the study. The parents of these children were then surveyed with the assistance of the kindergarten teachers. Before distributing the questionnaires, the classroom teachers utilized the time when parents were picking up their children to individually explain the purpose and significance of the survey. They clarified to the parents that the survey was solely for the research project and would not be used to evaluate the parents or the children. Additionally, parents were assured that the questionnaire would be filled out anonymously, would not reveal personal information about their children or themselves, and that verbal consent was obtained from all the parents surveyed.Between March 10 and April 10, 2023, our research team distributed a total of 258 questionnaires through the Questionnaire Star website. To ensure that parents felt comfortable answering the questionnaires, the importance of the questionnaire and the confidentiality of the responses were explained in the introductory section. After removing the questionnaires with identical answers for all items, 253 valid responses were recorded for the final data analysis.

### 2.2 Research tools

**2.2.1 Non-involvement scale.** The Non-intervention Strategy Scale developed by Jia lun Zhang (2010) [58] was used to examine whether parents adopt a non-intervention strategy to

handle sibling conflict. Taiwan and mainland China share a common cultural heritage, so the scale has a relatively strong cultural adaptation in mainland China. It has been widely used in related research in mainland China, with good reliability and validity testing [35,40]. The scale has 5 items. The participants rated themselves on a Likert scale that ranged from 1 (did not apply to me) to 5 (applied to me very much or most of the time). It includes items such as, "When he/she fights with their siblings, I will simply say 'stop it' and continue with my business without inquiring about their situation." In this study, Cronbach's α was 0.880.

**2.2.2 Sibling conflict scale.**   The parent-reported Sibling Conflict Questionnaire developed by Furman and Buhrmeste [59] was used as a tool to measure the sibling relationships of young children. The scale has been widely used in relevant studies both domestically and internationally, and it has undergone thorough testing for reliability and validity [60–62].The scale has 5 items. The scale is based on a Likert scale that ranges from 1 (not at all) to 5 (completely). It includes items such as, "He and his siblings yell at each other." Higher scores indicate that the child is less close to his or her siblings and that sibling conflict occurs more frequently. In this study, Cronbach's α was 0.812.

**2.2.3 Parent-child conflict scale.**   The Parent-Child Conflict Scale was developed by Pianta and Virginia [63]. The scale has been widely used in relevant studies both domestically and internationally, and it has undergone thorough testing for reliability and validity [64–66]. The scale has 11 items. It includes items such as, "He and I always seem to be fighting against each other." The scale is based on a 5-point system (1 means not at all, 5 means completely). The higher the score, the less intimate the relationship between the child and the parent, and the greater the conflict. In this study, Cronbach's α was 0.774.

**2.2.4 Social avoidance scale.**   The Social Avoidance scale developed by Sang [67] was used as the measurement tool. It has undergone thorough testing for reliability and validity. The scale was filled out by parents; therefore, the present study adopted the parents' perspective in formulating the questions. The scale has 4 items. It includes items such as, "He doesn't want to play with other children." The scale was scored on a 5-point scale, with 1 indicating a complete lack of conformity and 5 indicating complete conformity. Higher scores indicate children's heightened sense of social anxiety and a stronger inclination to avoid social situations. In this study, Cronbach's α was 0.745.

## 2.3 Data processing

The questionnaire data were analyzed using SPSS 21.0. The mediation effect test was conducted based on the method recommended by Zhonglin Wen and Baojuan Ye [68]. The data were organized and analyzed using Hayes' SPSS macro program Process [69].

## 2.4 Common method tests

Data collection through the questionnaire method needs to be tested for the presence of serious common method bias. Firstly, to ensure the scientific validity of the questionnaire data, a different scale format was used, such as changing the position of the independent and dependent variables in the questionnaire, to mitigate common method bias to some extent. Secondly, the questionnaire in this study was mainly parent-reported (as children in the early childhood stage could not complete the questionnaire). Therefore, Harman's one-way test was employed for the common method bias test [70], and the results indicated that the number of common factors with eigenvalues greater than 1 was 7. The percentage of variance explained by the first common factor was 25.118%, which was below the critical value of 50% [71]. Subsequently, the variance inflation factor (VIF) was evaluated, revealing that the VIF values of all constructs ranged between 1.294–1.401. According to Kock [72], a VIF value below 3.3

suggests that covariance is not a significant issue in the study model. Hence, there is no evidence of common method bias in this study.

# 3 Results

## 3.1 Data analysis

The descriptive statistics and correlation analysis of the variables are shown in Table 1. The table indicates that non-involvement strategies are significantly and positively correlated with sibling conflict, parent-child conflict, and social avoidance. Sibling conflict is significantly and positively correlated with parent-child conflict and social avoidance. Furthermore, parent-child conflict is significantly and positively correlated with social avoidance.

## 3.2 Analysis of inter mediation effects

The Process plug-in for SPSS software was used to conduct mediation effect analysis, selecting Model 6 and using a bootstrapping subsampling technique (5,000 times) to test the research hypotheses. The analysis aimed to examine the mediating roles of sibling conflict and parent-child conflict in the impact of non-intervention strategies on social avoidance in migrant children, while controlling for gender and age. Table 2 revealed that the non-intervention strategy directly and significantly positively predicted social avoidance ($\beta = 0.52$, $p < 0.001$). The non-intervention strategy significantly positively predicted sibling conflict ($\beta = 0.43$, $p < 0.001$). Sibling conflict ($\beta = 0.26$, $p < 0.001$) and non-intervention strategy ($\beta = 0.36$, $p < 0.001$) each significantly positively predicted parent-child conflict. Parent-child conflict ($\beta = 0.35$, $p < 0.001$) significantly positively predicted social avoidance, while the prediction of social avoidance by sibling conflict was not significant ($\beta = -0.07$, $p > 0.05$).

   The results of the analysis of the mediating effect showed (Table 3 and Fig 2) that parental non-intervention in sibling conflict had a direct effect on migrant children's social avoidance. The Bootstrap 95% confidence interval did not contain a value of 0, indicating that the direct effect reached a significant level, supporting H1. Sibling conflict and parent-child conflict partially mediated the relationship between parental non-intervention in sibling conflict strategy and migrant children's social avoidance, with a mediation effect value of 0.084, accounting for 26.5% of the total effect. Specifically, the mediating effect consisted of indirect effects generated by two pathways: indirect effect 1 (0.080) through the pathway of non-intervention strategy → parent-child conflict → social avoidance; and indirect effect 2 (0.022) through the pathway of non-intervention strategy → sibling conflict → parent-child conflict → social avoidance. Their Bootstrap 95% confidence intervals did not contain a value of 0, indicating that both indirect effects reached a significant level, supporting H3 and H4. In contrast, the Bootstrap 95% confidence interval for indirect effect 3 (-0.018) arising from the pathway of non-

**Table 1. Correlation analysis results for study variables.**

| Variable | M±SD | 1 | 2 | 3 |
|---|---|---|---|---|
| 1.Non-involvement Strategy | 13.45±6.02 | — | | |
| 2.Sibling Conflict | 17.07±4.32 | 0.42** | — | |
| 3.Parent-child Conflict | 27.21±9.25 | 0.48** | 0.40** | — |
| 4.Social Avoidance | 9.05±3.66 | 0.52** | 0.23** | 0.51** |

Note.

**p<0.01.

**Table 2. Regression analysis of variable relationships.**

| Regression Equation | | Integral Fit Integer | | | regression coefficient | significance |
|---|---|---|---|---|---|---|
| Outcome Variable | Predictor Variable | R | $R^2$ | F | β | t |
| Social Avoidance | Gender | 0.53 | 0.27 | 31.54 | -0.01 | -0.23 |
| | Age | | | | -0.06 | -1.02 |
| | Non-involvement Strategy | | | | 0.52 | 9.55** |
| Sibling Conflict | Gender | 0.43 | 0.17 | 18.57 | -0.06 | -1.04 |
| | Age | | | | 0.09 | 1.52 |
| | Non-involvement Strategy | | | | 0.43 | 7.40** |
| Parent-child Conflict | Gender | 0.54 | 0.28 | 25.61 | 0.04 | 0.71 |
| | Age | | | | -0.12 | -2.19 |
| | Non-involvement Strategy | | | | 0.36 | 5.98** |
| | Sibling Conflict | | | | 0.26 | 4.44** |
| Social Avoidance | Gender | 0.60 | 0.35 | 28.00 | -0.02 | -0.47 |
| | Age | | | | -0.02 | -0.30 |
| | Non-involvement Strategy | | | | -0.07 | -1.18 |
| | Sibling Conflict | | | | 0.35 | 5.77** |
| | Parent-child Conflict | 0.54 | 0.28 | 25.61 | 0.04 | 0.71 |

Note. **p<0.01.

intervention strategy → sibling conflict → social avoidance contains 0, indicating that this mediating effect is not significant, H2 is not supported.

## 4 Discussion

Based on previous empirical studies and related theories, this study examines the impact of parents' strategy of non-intervention in sibling conflict on migrant children's social avoidance and its mechanism of action. The findings hold theoretical and practical significance for advancing research on the relationship between parental non-intervention in sibling conflict and migrant children's social avoidance, as well as for enhancing migrant children's social adaptation.

### 4.1 The effect of non-intervention strategy on social avoidance in migrant children

It was found that parental non-intervention in sibling conflict positively predicts social avoidance in migrant children. This result validates H1, which suggests that parental non-intervention in conflicts between migrant children and their siblings leads to difficulties in social

**Table 3. Analysis of intermediation effects.**

| effect | pathway | effect value | Boot LLCI | Boot ULCI |
|---|---|---|---|---|
| direct effect | Direct pathway:Non-involvement Strategy →Social Avoidance | 0.233 | 0.160 | 0.300 |
| indirect effect | 1.Non-involvement Strategy → Parent-child Conflict→Social Avoidance | 0.080 | 0.047 | 0.124 |
| | 2.Non-involvement Strategy →Sibling Conflict→Parent-child Conflict→Social Avoidance | 0.022 | 0.010 | 0.039 |
| | 3.Non-involvement Strategy→ Sibling Conflict→Social Avoidance | -0.018 | -0.045 | 0.010 |
| Total indirect effect | | 0.084 | 0.040 | 0.130 |

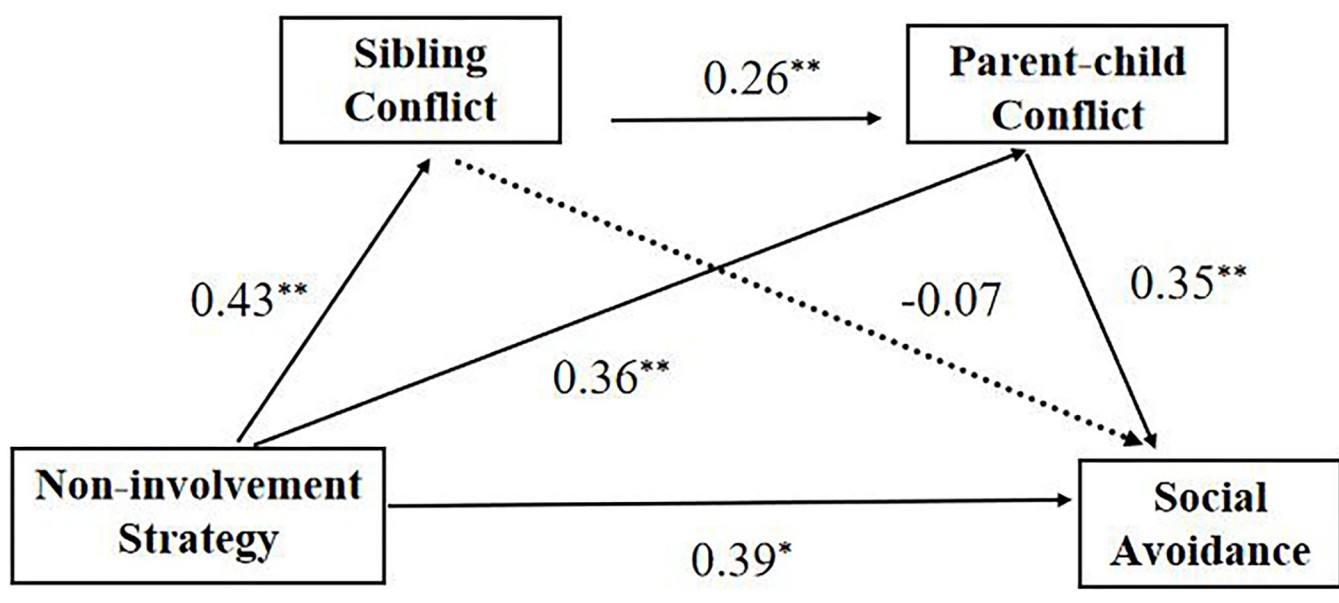

**Fig 2. Chain mediation effect diagram.**

adjustment for the children, as evidenced by their avoidance of social interactions. This finding is consistent with previous research [19,20,73]. Migrant children are in a critical period of socialization, and their new living environment presents specific challenges to their socialization process. The family, as the "micro-system" of individual growth, plays an irreplaceable role in the socialization process of children [74].

When children experience sibling conflict and parents adopt a strategy of non-intervention, failing to provide help and support to the child, the individual may begin to doubt themselves and the world around them. This lack of support can lead to unmet psychological needs, ultimately resulting in avoidance of peer interactions, anxiety, and a range of emotional problems such as depression and loneliness [75]. At the same time, parenting styles serve as a crucial source of social support for individuals as they mature [76]. These styles are closely linked to children's interpersonal and pro-social adjustment, which can either foster or hinder individual interpersonal and social adaptation [77].

The study also found that parental non-intervention in sibling conflict remained a direct predictor of social avoidance in mobile children even after considering mediating variables. This further highlights the significant role of parents in children's social development, aligning with the spillover hypothesis of family systems theory [78]. Families and parents are important sources of social support for preschool children as they grow up. Children tend to accept their parents' parenting styles and eagerly anticipate forming strong bonds with them. Children's behaviors further solidify the impact of their parents' parenting styles.

### 4.2 The mediating role of parent-child conflict

The present study found that parental non-intervention in sibling conflict strategies influences migrant children's social avoidance through parent-child conflict. This reveals a mediating mechanism by which parental non-intervention in sibling conflict strategies impacts migrant children's social avoidance, validating H 3. When a conflict arises between a child and a sibling, a parent's strategy of non-intervention, often ignoring the child's needs and showing a lack of sensitivity and concern for the child, can result in a lack of emotional support for the

child. This behavior can cause coldness and indifference in the parent-child relationship, which is consistent with existing research [79]. Parent-child attachment theory suggests that children develop an internal working model based on their early experiences of interacting with their caregivers. This internal working model consists of a self-model and a parental model, with the parental model containing three factors: trustworthiness, sensitivity, and caring [80] In a conflictual parent-child relationship where children do not receive trust, care, and support from their parents, it is difficult for children to develop a sense of trust. Consequently, it becomes harder for them to socialize with others, and their ability to explore, comply, integrate, and adapt in urban settings is compromised. This aligns with previous research findings [81,82]. When migrant children have conflicts with their peers, it is challenging for them to handle the situations. Unfortunately, their parents often show neglect and refrain from intervening, making it hard for the children to receive trust, care, and support from them. The dissatisfaction with their peers can lead to negative psychological impacts on migrant children, causing them to exhibit low adaptability and a tendency to avoid social interactions. This phenomenon has been supported by studies in China on the urban adaptation of migrant children [83].

### 4.3 Chain mediation of sibling conflict and parent-child conflict

It was found that the chain mediation constituted by sibling conflict and parent-child conflict is also an important pathway through which parental nonintervention in sibling strategies affects the development of social avoidance in migrant children. This result validates H 4 and supports family systems theory [32,79,84–86]. By not intervening in conflicts between their children, parents miss out on teaching communication skills to young children who lack appropriate sibling conflict resolution. Consequently, when sibling conflicts reoccur again, children may escalate the conflicts due to a lack of proper conflict resolution, which is consistent with existing findings [28]. Sibling quarrels and disagreements cause worry and distress for parents. These conflicts also impact parents' attitudes toward their children, subsequently affecting the overall climate and emotional tone of the parent-child relationship [87], resulting in a conflicted parent-child relationship. It has been found that prolonged parent-child conflict can lead to numerous socially problematic behaviors in individuals, such as negative social adjustment, which was also confirmed in this study [54]. Thus, the mediating role of sibling conflict and parent-child conflict connects sibling and parent-child relationships within the family system. This suggests that a non-involvement strategy, a neglectful parental approach to handling sibling conflict, can impact the child's social development. It can result in a negative social interaction experience, ultimately leading to social avoidance.

This chain mediation role integrates various elements of the family relationship system, focusing on the social adaptation of migrant children in a new environment. It reveals a strong interconnection among the elements within the system, aligning with prior research [85,86]. However, this study differs in its emphasis on the sibling relationship as a mediator of the second variable, exploring how sibling relationships impact parent-child relationships. This aspect has received limited attention in existing studies, making it a significant contribution of this research.

### 4.4 The mediating role of sibling conflict

The present study found that sibling conflict did not significantly mediate between parental nonintervention in sibling conflict and social avoidance in mobile children, a result that is inconsistent with H 2. Although parental nonintervention in sibling conflict significantly and positively predicted sibling conflict, the direct effect of sibling conflict on social avoidance in

mobile children was not significant. This finding prompts us to consider the positive effects of sibling conflict. It has been found that the more time children spend with their siblings, the more sibling conflict they have [16]. Compatriots in conflict have ample opportunities to enhance their communication skills without jeopardizing their relationship [88,89]. It has been found that sibling conflicts can be categorized into constructive conflicts and destructive conflicts based on how they affect individuals. Constructive conflicts, which involve controlling emotions, maintaining social interactions, and solving problems fairly through negotiation and reasoning, can help develop children's communication skills. This is crucial for children's development and socialization [90,91]. Therefore, the revelation of this study is that parents need to face up to sibling conflicts and not simply stop the conflict between the children in a rough way, but correctly guide their children effectively to resolve these conflicts.

## 5 Research value

Based on family systems theory and ecological systems theory, this study examines the impact of parents' non-intervention strategy in sibling conflict on migrant children's social avoidance. It also explores the mediating role of sibling conflict and parent-child conflict. The study's theoretical significance lies in enhancing research on family relationships, particularly by illuminating the chain mediating role of sibling conflict and parent-child conflict. This sheds light on the influences and constraints within the family relationship system. Importantly, the study refines the research on the effects of sibling conflict, suggesting that we should consider these effects from a dialectical perspective. The practical value of this study is to offer parents concrete guidance on actively managing family relationships within the new demographic reproduction system. It also aims to help migrant children better integrate into society by providing scientific guidance on fostering positive sibling relationships and nurturing close parent-child bonds.

## 6 Research shortcomings and prospects

First, the subjects of this study are preschool children, who are in an important period of socialization. Behavioral problems may change with age. This study is a cross-sectional study, and a longitudinal study can be adopted in the future to explore the mechanism of action in more depth. Second, the examination of parental strategies for handling sibling conflict has primarily focused on the family system and has not considered the influence of other factors. Studies have pointed out that the ways in which parents intervene in sibling conflict are related to their own gender [92–94] Additionally, the way in which parents intervene in sibling conflict changes as their children get older [95]. Therefore, future research will also consider whether there are other factors that influence the way in which parents intervene in sibling conflict. Third, this study focuses on non-intervention strategies, how child-centered and control strategies [17,31] impact the social development of mobile children. What about the mediating role of sibling and parent-child relationships. These aspects could be crucial directions and components for future research. Fourth, as the subject of this study is migrant children, do the elements of the family relationship system affect the social development of local children in the same way as that of migrant children. What is the mechanism of influence. These aspects could be crucial directions and components for future research.

## Supporting information

**S1 Data. Original empirical study data.**
(XLSX)

**S1 Appendix. A constructs, measurement items and sources.**
(DOCX)

## Author Contributions

**Investigation:** Danping Ye.

**Writing – original draft:** Yuge Yue.

**Writing – review & editing:** Kaiwu Lu.

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
