## [Decision Letter · Decision Letter 0]

9 Jul 2024

PONE-D-24-12364Parental Non-involvement Strategy for Handling Sibling Conflict on Social Avoidance in Migrant Children: Chain Mediation of Sibling Conflict and Parent-Child ConflictPLOS ONE

Dear Dr. Lu,

Thank you for submitting your manuscript to PLOS ONE. After careful consideration, we feel that it has merit but does not fully meet PLOS ONE’s publication criteria as it currently stands. Therefore, we invite you to submit a revised version of the manuscript that addresses the points raised during the review process.

We look forward to receiving your revised manuscript.

Kind regards,

Muddsar Hameed

Academic Editor

PLOS ONE

“This study was supported by a grant from the Office for Philosophy and Social Sciences in Fujian Province, China (Title: Examining the New Era's Urban Floating Population's Family Relationships in Fujian Province. ID: FJ2022X022)”

Reviewers' comments:

Reviewer's Responses to Questions

**Comments to the Author**

1. Is the manuscript technically sound, and do the data support the conclusions?

Reviewer #1: Partly

Reviewer #2: Yes

2. Has the statistical analysis been performed appropriately and rigorously? 

Reviewer #1: Yes

Reviewer #2: Yes

3. Have the authors made all data underlying the findings in their manuscript fully available?

Reviewer #1: Yes

Reviewer #2: Yes

4. Is the manuscript presented in an intelligible fashion and written in standard English?

Reviewer #1: Yes

Reviewer #2: Yes

5. Review Comments to the Author

Reviewer #1: Work on the article, literature, hypothesis, tools. Moreover also update the article with recent knowledge. Improve the limitations and recommendations of this article. To improve the quality and content of your article.

Reviewer #2: • The reference for the Bronfenbrenner's ecological systems theory has not been provided

• Provide China's family model reference

• Define key terms such as "social avoidance" and "social adjustment" upfront to ensure readers understand these concepts from the beginning

• Further explanation of the mechanisms by which parent-child conflict and sibling conflict influence social avoidance would add depth to the analysis.

• Ensure all citations follow a consistent format. There are some inconsistencies in citation styles and formatting

• The use of convenience sampling may limit the generalizability of the findings whereas random sampling technique enhances the representativeness of the sample, given the sample size is small.

• The study relies heavily on self-reported data from parents, which can introduce response biases.

• The study does not include the perspectives of the children themselves.

6. PLOS authors have the option to publish the peer review history of their article (what does this mean?). If published, this will include your full peer review and any attached files.

Reviewer #1: No

Reviewer #2: **Yes: **Khazeneen Atiq

---

## [Author Response · Author response to Decision Letter 0]

24 Jul 2024

Dear reviewers and editors：

We would like to thank you for your careful reading, helpful comments, and constructive suggestions, which has significantly improved the presentation of our manuscript.

We have carefully considered all comments from the reviewers and revised our manuscript accordingly. In the following section, we summarize our responses to each comment from the reviewers (See red font in the text for details of the changes). We hope our revised manuscript can be accepted for publication.

The following is a point-by-point response to the editors’ comments.

1.Abstract Clarity: The abstract provides a good overview but can be improved for clarity. Consider explicitly stating the research design, key findings, and implications in a structured manner (e.g., Objectives, Methods, Results, Conclusions).

Response 1: We are grateful for the suggestion. We have revised it in the abstracts using red typeface. I am truly grateful for all your comments. I was deeply moved by your help.

In the process of urbanization, the social adaptation of migrant children has become an important issue in their development. This study adopts family systems theory and ecological systems theory to examine the effects of parental non-involvement strategies in handling sibling conflict on migrant children's social avoidance. It also investigates the mediating role of sibling conflict and parent-child conflict. The results of the study, reported by parents of 253 mobile children with siblings, suggest that parental strategies of not intervening in sibling conflict are an important factor influencing the development of social avoidance in mobile children. Parental strategy of not intervening in sibling conflict had an effect on migrant children's social avoidance through the separate mediating effect of parent-child conflict, and also through the chained mediating effect of sibling conflict and parent-child conflict. The study also found that the separate mediating effect of sibling conflict was not significant. This study contributes to the research on the relationship between parental non-intervention in sibling conflict and migrant children's social avoidance. It also highlights the impact of sibling conflict and parent-child conflict on migrant children's social avoidance by establishing and validating a comprehensive research model. The results of the study can help parents establish close parent-child relationships for migrant children and provide scientific guidance for children to develop positive sibling relationships. This, in turn, can assist migrant children in better adapting to new social environment.

2.Literature Review: The introduction could benefit from a more comprehensive literature review. Incorporate recent studies on sibling conflict, parental non-involvement, and social avoidance to establish a stronger theoretical foundation.

Response 2: We are grateful for the suggestion. We have revised it in the introduction using red typeface. I am truly grateful for all your comments, which make the introduction establish a stronger theoretical foundation.

3.Hypotheses Statement: Clearly state your research hypotheses at the end of the introduction. This helps in guiding the reader and setting clear expectations for the study.

Response 3: We are grateful for the suggestion. We have revised it at the end of the introduction using red typeface. I am truly grateful for all your comments, which help the reader establish a clear expectations for the study.

4.Sample Size Justification: Provide a justification for the sample size of 253 children. Include a power analysis to demonstrate that the sample size is adequate to detect the expected effects.

Data Collection Method: Detail the data collection process, including how participants were recruited and how the surveys were administered. Mention any ethical considerations and how informed consent was obtained.

Ethical Considerations: Expand on the ethical considerations in the methodology section. Mention the ethical review board's approval and any measures taken to protect participants' confidentiality and well-being.

Response 4: We are grateful for the suggestion. We have revised it in the Data collection using red typeface. I am truly grateful for all your comments. Your suggestions make our sample size and data collection more scientific.

After receiving approval from the Research Ethics Review Committee of Ningde Normal University , this study used convenience sampling to select 10 kindergartens in the four cities of Xiamen, Zhangzhou, Quanzhou, and Ningde in China Fujian Province. Subsequently, 30 children with siblings were randomly chosen from each kindergarten, totaling 300 children. With the assistance of kindergarten teachers, the children's birthplaces were verified, leading to the exclusion of 42 local children. Finally, 258 migrant children were included in the study. The parents of these children were then surveyed with the assistance of the kindergarten teachers. Before distributing the questionnaires, the classroom teachers utilized the time when parents were picking up their children to individually explain the purpose and significance of the survey. They clarified to the parents that the survey was solely for the research project and would not be used to evaluate the parents or the children. Additionally, parents were assured that the questionnaire would be filled out anonymously, would not reveal personal information about their children or themselves, and that verbal consent was obtained from all the parents surveyed.Between March 10 and April 10, 2023, our research team distributed a total of 258 questionnaires through the Questionnaire Star website. To ensure that parents felt comfortable answering the questionnaires, the importance of the questionnaire and the confidentiality of the responses were explained in the introductory section. After removing the questionnaires with identical answers for all items, 253 valid responses were recorded for the final data analysis.

5.Measurement Tools: Provide more details on the psychometric properties of the measurement tools used (e.g., validity, reliability). Mention any cultural adaptations made for the scales used in the Chinese context.

Response 5: We are grateful for the suggestion. We have revised it in the 2.2Research tools using red typeface. I am truly grateful for all your comments. We have added evidence of the reliability and validity of the scale for use in Chinese and international studies.

2.2 Research tools

2.2.1 Non-involvement Scale

Taiwan and mainland China share a common cultural heritage, so the scale has a relatively strong cultural adaptation in mainland China. It has been widely used in related research in mainland China, with good reliability and validity testing [36,41]. 

2.2.2 Sibling Conflict Scale

The scale has been widely used in relevant studies both domestically and internationally, and it has undergone thorough testing for reliability and validity[61,62,63].

2.2.3 Parent-Child Conflict Scale

The scale has been widely used in relevant studies both domestically and internationally, and it has undergone thorough testing for reliability and validity[65,66,67]. 

2.2.4 Social Avoidance Scale

The Social Avoidance scale developed by Sang [68] was used as the measurement tool. It has undergone thorough testing for reliability and validity. 

6.Statistical Analysis: Explain the choice of statistical methods in more detail. For example, discuss why the SPSS macro developed by Hayes was chosen for mediation analysis and how the results were interpreted.

Response 6: We are grateful for the suggestion. We have revised it in the 2.3 Data Processing using red typeface. I am truly grateful for all your comments. Your suggestions make our study more scientific.

2.3 Data processing

The questionnaire data were analyzed using SPSS 21.0. The mediation effect test was conducted based on the method recommended by Zhonglin Wen and Baojuan Ye[69]. The data were organized and analyzed using Hayes' SPSS macro program PROCESS[70].

7.Common Method Bias: The study mentions using Harman's one-way test for common method bias. Consider discussing the potential limitations of this test and explore additional methods to mitigate common method bias.

Response 7: We are grateful for the suggestion. We have revised it in the 2.4 Common method tests using red typeface. I am truly grateful for all your comments. Your suggestions make our study more scientific.

2.4 Common method tests

Data collection through the questionnaire method needs to be tested for the presence of serious common method bias. Firstly, to ensure the scientific validity of the questionnaire data, a different scale format was used, such as changing the position of the independent and dependent variables in the questionnaire, to mitigate common method bias to some extent. Secondly, the questionnaire in this study was mainly parent-reported (as children in the early childhood stage could not complete the questionnaire). Therefore, Harman's one-way test was employed for the common method bias test [71], and the results indicated that the number of common factors with eigenvalues greater than 1 was 7. The percentage of variance explained by the first common factor was 25.118%, which was below the critical value of 50%[72]. Subsequently, the variance inflation factor (VIF) was evaluated, revealing that the VIF values of all constructs ranged between 1.294-1.401. According to Kock[73], a VIF value below 3.3 suggests that covariance is not a significant issue in the study model. Hence, there is no evidence of common method bias in this study.

8.Discussion Depth: The discussion section should be more comprehensive. Relate your findings to existing literature, explain possible reasons for your findings, and discuss the practical implications for parents and educators.

Figures and Tables: Ensure that all figures and tables are clearly labeled and provide additional context where necessary. For example, Figure 1 should be referenced in the text with a brief explanation of its contents.

Response 8: We are grateful for the suggestion. We have revised it in the 3 Results, 4 Discussion, and 5 Research value using red typeface. I am truly grateful for all your comments. Your suggestions make our study more clearer and scientific.

9.Limitations and Future Research: While some limitations are mentioned, provide a more thorough discussion. Suggest specific areas for future research, such as longitudinal studies or the impact of different parental strategies on sibling conflict resolution.

Response 9: We are grateful for the suggestion. We have revised it in the 6 Research shortcomings and prospects using red typeface. I am truly grateful for all your comments. We provide a more thorouge discussion

First, the subjects of this study are preschool children, who are in an important period of socialization. Behavioral problems may change with age. This study is a cross-sectional study, and a longitudinal study can be adopted in the future to explore the mechanism of action in more depth.

Third, this study focuses on non-intervention strategies, how child-centered and control strategies [32,18] impact the social development of mobile children? What about the mediating role of sibling and parent-child relationships? These aspects could be crucial directions and components for future research. 

Fourth, as the subject of this study is migrant children, do the elements of the family relationship system affect the social development of local children in the same way as that of migrant children? What is the mechanism of influence? These aspects could be crucial directions and components for future research.

10.References: Update the references section to include the most recent studies and ensure all citations are properly formatted according to the journal's guidelines.

Response 10: We are grateful for the suggestion. We have update the references, including the most recent studies and the citations format according to the juurnal’s guidelines. I am truly grateful for all your comments. 

11.Language and Style: Ensure the manuscript is free from grammatical errors and uses clear, concise language. Consider professional editing to enhance readability.

Response 11: We are grateful for the suggestion. We have revised the Language and Style. I am truly grateful for all your comments. 

12.Supplementary Materials: If applicable, provide supplementary materials such as survey questionnaires or detailed statistical outputs to support the findings.

Response 12: We are grateful for the suggestion. Research materials can be obtained from the corresponding author if required.

Once again, we thank the reviewers and editors for recognizing our research! We sincerely hope this manuscript will be finally acceptable to be published . Thank you very much for all your help and looking forward to hearing from you soon. 

Best regards 

Sincerely yours

---

## [Editor Report · Decision Letter 1]

26 Jul 2024

Parental Non-involvement Strategy for Handling Sibling Conflict on Social Avoidance in Migrant Children: Chain Mediation of Sibling Conflict and Parent-Child Conflict

PONE-D-24-12364R1

Dear Dr. Kaiwu Lu ,

We’re pleased to inform you that your manuscript has been judged scientifically suitable for publication and will be formally accepted for publication once it meets all outstanding technical requirements.

Kind regards,

Muddsar Hameed

Academic Editor

PLOS ONE

---

## [Editor Report · Acceptance letter]

31 Aug 2024

PONE-D-24-12364R1 

PLOS ONE

Dear Dr. Lu, 

I'm pleased to inform you that your manuscript has been deemed suitable for publication in PLOS ONE. Congratulations! Your manuscript is now being handed over to our production team.

Kind regards, 

on behalf of

Dr. Muddsar Hameed 

Academic Editor

PLOS ONE